# Extraction and Analysis of Respiratory Motion Using a Comprehensive Wearable Health Monitoring System

**DOI:** 10.3390/s21041393

**Published:** 2021-02-17

**Authors:** Uduak Z. George, Kee S. Moon, Sung Q. Lee

**Affiliations:** 1Department of Mathematics and Statistics, San Diego State University, San Diego, CA 92182, USA; ugeorge@sdsu.edu; 2Department of Mechanical Engineering, San Diego State University, San Diego, CA 92182, USA; 3Electronics and Telecommunications Research Institute, Daejeon 34129, Korea; hermann@etri.re.kr

**Keywords:** biomedical signal processing, wearable biomedical sensors, medical equipment, multi-sensor fusion, respiration, tidal volume, cubic spline interpolation

## Abstract

Respiratory activity is an important vital sign of life that can indicate health status. Diseases such as bronchitis, emphysema, pneumonia and coronavirus cause respiratory disorders that affect the respiratory systems. Typically, the diagnosis of these diseases is facilitated by pulmonary auscultation using a stethoscope. We present a new attempt to develop a lightweight, comprehensive wearable sensor system to monitor respiration using a multi-sensor approach. We employed new wearable sensor technology using a novel integration of acoustics and biopotentials to monitor various vital signs on two volunteers. In this study, a new method to monitor lung function, such as respiration rate and tidal volume, is presented using the multi-sensor approach. Using the new sensor, we obtained lung sound, electrocardiogram (ECG), and electromyogram (EMG) measurements at the external intercostal muscles (EIM) and at the diaphragm during breathing cycles with 500 mL, 625 mL, 750 mL, 875 mL, and 1000 mL tidal volume. The tidal volumes were controlled with a spirometer. The duration of each breathing cycle was 8 s and was timed using a metronome. For each of the different tidal volumes, the EMG data was plotted against time and the area under the curve (AUC) was calculated. The AUC calculated from EMG data obtained at the diaphragm and EIM represent the expansion of the diaphragm and EIM respectively. AUC obtained from EMG data collected at the diaphragm had a lower variance between samples per tidal volume compared to those monitored at the EIM. Using cubic spline interpolation, we built a model for computing tidal volume from EMG data at the diaphragm. Our findings show that the new sensor can be used to measure respiration rate and variations thereof and holds potential to estimate tidal lung volume from EMG measurements obtained from the diaphragm.

## 1. Introduction

Reliable unobtrusive monitoring of respiration is of great importance in critically ill patients and ordinary healthy people. Several research groups have reported various respiration monitoring methods. However, we need to overcome some technical challenges to develop genuinely wearable sensors that can become practical and clinically meaningful. Particularly, continuous respiratory monitoring requires to address the technical issues of battery power source, sensor data storage, wireless data communication, and automated diagnosis [1,2]. There have been significant innovations to achieve wearability using a range of material properties, structure, and integration. For example, stretchable sensors, such as strain gauges and many novel materials embedded bands, were attached to the human chest to measure local strain realizing the respiration monitoring [3,4,5]. Bioimpedance devices have been used for measuring lung capacity [6,7] because bioimpedance has a linear relationship with respiratory volume during normal breathing. In other studies, the inertial measurement units (IMU) were attached to the abdomen and chest to monitor respiratory behaviors [8,9,10]. Also, recently skin-mounted soft electronics were reported to detect the human motions toward motion recognition [11,12,13].

Understanding respiration sound characteristics has been one of the earliest and popular methods of detecting early respiratory illnesses with abnormal lung sounds [14,15,16,17]. It is understood that the coordinated contraction of respiratory muscles such as the diaphragm and external intercostals increases the ribcage and the chest’s rising [18,19,20,21,22]. For example, the diaphragm, which, during regular inspiration, contracts and flattens, pushing on the abdomen, while the lower ribs move upwards and outwards. The muscle movements of the diaphragm and the ribcage induce air flow through the trachea and bronchi. It is the flow of air that makes the sound signals. It is also well-known for many years that the respiration cycles and heartbeats are tightly coupled [23,24,25,26,27]. For example, the respiratory motion can also be estimated by analysis of electrocardiogram (ECG) variations. The heart rate increases during inspiration; during expiration, it decreases again [16]. Thus, an integrated approach of using respiratory sounds and cardiovascular physiological effects simultaneously can help model the breathing patterns, such as respiratory rate, tidal volume, and diaphragmatic activation, more precisely.

In this study, we have built a light-weight wireless wearable digital health lung sensor system. This paper presents a novel way to monitor lung health by using a stethoscope and EMG integrated wearable sensors ready to employ artificial intelligence to detect lung function changes remotely. This paper is organized as follows. Section 2 describes a custom-designed multi-channel integrated sensor system to detect lung sounds and muscle activity simultaneously. The multi-sensor device tracks cardiogram and breathing while sticking to human skin comfortably. The system can measure lung function with a sound transducer and a set of EMG electrodes remotely. The whole system weighs only 15 g and sticks to human skin like a Band-Aid. Further, it can synthesize the sensor data from very different sources (e.g., EMG and sound data) for estimating the comprehensive state of lung health.

Section 3 describes the study to build tidal volume estimation models to reproduce lung mechanics using mathematical methods and statistical analysis. It provides a unique way of estimating the tidal volume using high-frequency EMG signals from breathing muscles (e.g., diaphragm and the external intercostal muscles). Section 4 introduces a new signature matrix-based signal processing method suitable for sensor fusion and scalable machine learning algorithms. The new approach can integrate sensor data with varying reliability and sensitivity using new statistical feature extraction. The proposed wearable sensor system is suitable for daily use and to engage in digitally interactive lung function monitoring. The device and the computer program were developed and demonstrated.

## 2. Wearable Sensor System Overview

The development of the electronic and digital wearable health sensors can introduce computer-aided monitoring and diagnosis by automated analysis, graphic visualization, storage, and archiving. Figure 1 and Figure 2 show the wireless wearable health sensors, which provide the capability to monitor and record the sound and the biopotential signals of the heart and the lung remotely onto their PC or laptop for further visualization and analysis. The system has sensor electrodes and transducer, the data acquisition and signal-processing circuit, wireless data transmission chip in the wearable sensor system. The analysis and diagnosis of the transmitted signals are conducted in the external computer system. The output of the signal processing in the external computer system is the feature classification result for clinical diagnostic decision making. Detailed descriptions of the sensor system components and their specifications are provided in Table 1.

The sound detection is achieved in the wearable sensor unit by placing a piezoelectric microphone mounted on the sensor’s middle. A piezoelectric transducer converts the heart and lung sound signals to analog electrical signals in the stethoscope. Furthermore, the system can transmit heart sounds wirelessly up to 10 m to a remote processing network, promoting telemedicine’s evolution and potential applications. It also allows the possibility of automatic acoustic interpretation in cardiovascular and respiration diagnostics. The sound acquisition module uses a piezoelectric microphone with a sampling frequency of 4 kHz. An INTAN chip-based analog-to-digital conversion with a 16-bit resolution and pre-amplification is performed inside the wearable sensor unit. The raw sound signals include cardiac sounds with a spectrum 20–100 Hz.

The wearable sensor also comprises of multiple channels (8-ch) that communicate wirelessly using Bluetooth low-energy (BLE) technology, as shown in Figure 2. For example, the biopotential electrodes for EMG sensors acquire the heartbeat and breathing muscle activation signals and feed it to the analog-font-end, for pre-amplification. After that, the signal is transmitted wirelessly into a personal computer (PC), where the signal data is processed and classified using MATLAB. The sensor system accomplishes a real-time multiple sensor signal acquisition, amplification, filtering, digitization, and wireless transmission. For this study, a custom wearable sensor was designed and implemented using two commercial adhesive EMG patches, as shown in Figure 1.

## 3. Experimental Results

In the study, a set of experimental sensor signals was collected from a healthy male and female with slow respiration with a constant rate (i.e., 4-s inspiration and 4-s expiration). The breathing cycle was controlled using a metronome (https://www.imusic-school.com/en/tools/online-metronome/, accessed on 31 December 2020). Five different tidal volumes (i.e., 1000, 875, 750, 625 and 500 mL) were controlled by using a Voldyne 5000 Spirometer (Hudson RCI, calibrated with PF100 digital Peak Flow & FEV1 Meter, Microlife, Clearwater, FL, USA). The wearable system has a bandpass filter program installed to maximize the sensor’s heart and lung signals. The sampling frequency was set by 4 kHz per channel for the experiment. A higher sampling rate can also be arranged for greater accuracy at more power consumption.

Figure 3 represents the time and amplitude characteristics of typical lung sound and EMG signals obtained from the diaphragm muscle location. The first 4-s represents the inspiration phase (shown in a red box), and the following 4-s one the expiration. The sound and EMG signals in the figure show synchronized patterns in the graphs since the respiration cycles, and heartbeats are tightly coupled [23,24,25,26,27] (see purple and orange circles in red boxes). Further, in the figure, the heartbeat signal heights variations during the inspiration and expiration respiratory motions are noticeable (blue circles). It is known that the heartbeat signal heights are shortened during inspiration [28].

Figure 4a shows the filtered signals to separate the low and high-frequency lung sound signals (the first and the second figures) and the low and high-frequency EMG signals obtained from the diaphragm muscle location (the third and the fourth figures) using the proposed multimodal wearable sensor. It is clear from Figure 4a that the heartbeat and cardio sounds can be monitored simultaneously. Further, as noted from Figure 3, the first (sound under 100 Hz) and the third (EMG under 50 Hz) figures show the typical heartbeat signals in synchrony with the phases of respiration (purple circle), whereby heartbeat signal heights are shortened during inspiration (blue circle) and lengthened during expiration [29]. From the fourth figure and Figure 3, it can also be noted that there is clear evidence of the increasing intensity of high-frequency EMG signals at the diaphragm muscle location during the phase of inspirations (orange circles) compared to expirations.

Problems in either heart valves or the heart muscles result in abnormal heart sounds, and murmurs can be accounted for other cardiovascular issues [30]. The second figure shows that the proposed wearable sensor can monitor. Figure 4b shows one example of the stethoscope audio signals obtained from diaphragm muscle location (1000 mL tidal volume; 2-s inspiration (red box)). The sound signals were recorded using a conventional commercial stethoscope head (shown in Figure 4b picture) with an audio microphone (30 dB sensitivity, 20–20,000 Hz range, 20 kHz sampling rate). We used a stethoscope diaphragm to pick the lung and heart sounds from the body. The QuickTime software allowed us to directly record the sound onto a computer for further visualization and analysis using MATLAB. Figure 4b also shows the heart sounds (red circles), which we can find from Figure 4a with the wearable sensor device.

For the validation, we also used a Delsys Trigno Wireless EMG system (Delsys Inc., Natick, MA, USA) as a reference system to compare with the electromyographs from the wearable sensor. It is a widespread commercial EMG system with 16 channels and a sampling rate of 2000 Hz. The signal acquisition was carried out via the EMGworks software. Figure 4c shows the reference EMG signals obtained from diaphragm muscle location. Both Figure 4a,b show heartbeat signal height variation as well as increasing intensity of high-frequency EMG signals at the diaphragm muscle location during the phase of inspirations (orange circles).

## 4. Modeling of Respiration Function

This section describes the methods for modeling respiration functions, such as the respiration rate and the tidal volume from the wearable sensor signals.

### 4.1. Extraction of Respiration Volume

#### 4.1.1. Cubic Spline Interpolation

Cubic spline interpolation is a type of interpolation method that involves the approximation of data points by piecewise cubic functions/splines [31]. Each cubic function connects adjacent data points and is useful for estimating the value of a function within the range of a discrete set of known data points without knowing the actual function. Using cubic spline interpolation method, we derived a function that estimates the tidal volume from the area under the curve (AUC) of a given EMG graph (see Figure 5). To accomplish this, we create a cubic function *S_j_* that connects adjacent points of tidal volumes and corresponding AUC per breathing cycle:(1)$$S_{j}\left(x\right)=a_{j}{\left(x-x_{j}\right)}^{3}+{\;b}_{j}{\left(x-x_{j}\right)}^{2}+{\;c}_{j}\left(x-x_{j}\right)+d_{j}\;for\;j=0,\ldots ,n-1$$
where n denotes the number of data points and $x_{j}$ denotes known mean AUC for the high-frequency EMGs per breathing cycle (see Figure 5) collected by the sensor at either the diaphragm or EIM location. $S_{j}\left(x\right)$ is the $j^{th}$ cubic spline describing the tidal volume as a function of *x*. The cubic splines $S_{j}\left(x\right)$ connects adjacent data points therefore the number of splines will be one less than the number of data points as stated in Equation (1). Relevant conditions are required in order to determine the coefficients $a_{j},\;b_{j},\;c_{j},$ and $d_{j}$ of the cubic splines. We impose that the cubic splines must match the function values at known points and that the first and second derivatives of adjacent splines are continuous [20]. This gives us the following 4*n* − 2 conditions:(2)$$S_{j}\left(x_{j}\right)=y_{j}\;for\;j=0,\ldots ,n-1$$
(3)$$\;S_{n-1}\left(x_{n}\right)=y_{n}\;for\;j=0,\ldots ,n-1$$
(4)$$S_{j}\left(x_{j+1}\right)=S_{j+1}\left(x_{j+1}\right)j=0\;for\;j=0,\ldots ,n-2$$
(5)$$S_{j}^{\prime }\left(x_{j+1}\right)=S_{j+1}^{\prime }\left(x_{j+1}\right)\;for\;j=0,\ldots ,n-2$$
(6)$$S_{j}^{\prime \prime }\left(x_{j+1}\right)=S_{j+1}^{\prime \prime }\left(x_{j+1}\right)\;for\;j=0,\ldots ,n-2\;$$
where $y_{j}$ are known values of tidal volume corresponding to the mean AUC for the high-frequency EMGs per breathing cycle. Equation (1) has 4n undetermined coefficients, therefore we need two more conditions in order to obtain the 4n conditions necessary to determine the coefficients. We set two additional conditions by imposing natural boundary conditions, i.e.,
(7)$${S_{0}}^{\prime \prime }\left(x_{0}\right)=0$$
(8)$${S_{n-1}}^{\prime \prime }\left(x_{n}\right)=0$$
and corresponds to setting the curvature to zero at the end points. Using the conditions Equations (2)–(6) and Equations (7)–(8) we are able to solve for the coefficients $a_{j},\;b_{j},\;c_{j},$ and $d_{j}$ in Equation (1).

Figure 6 shows box plots of the AUCs during one breathing cycle for different tidal volumes for two different people. Using the spline interpolation method, we can predict the tidal volume using the AUC calculation for the EMG data collected from the sensor. The high-frequency EMG data collected at the EIM location have somewhat higher variations than those from the diaphragm location (Figure 6). Thus, it appears that the diaphragm sensor placement looks more appropriate for tidal volume estimation. Diaphragm and EIM are the vital muscle for respiration, requiring both for inspiration in the form of muscle contraction and expiration by relaxation. In our study, the AUC range for high-frequency EMG data obtained at the EIM is smaller than those obtained at the diaphragm and is consistent with what we would expect for the data since the EIM are a group of 11 pairs of tiny muscles found between the ribs. The sensor was placed on only one of the 11 pairs of EIM; thus, the EMG intensity will be lesser. During inspiration, it is known that the diaphragm movement displaced about 700 mL, and the right hemidiaphragm dome shortened by about 70 mm [32]. Further, the high-frequency filtered EMG signals (>50 Hz) can reduce unwanted low-frequency EMG artifacts from body motion.

#### 4.1.2. Analysis of Variance (ANOVA)

The respiratory volume is extracted by computing the area under the EMG graph per breathing cycle. The box plot in Figure 6 visually presents the area under the curve (AUC) for different tidal volume at the external intercostal muscle (EIM) and diaphragm. The null hypothesis *H*_0_ is that there is no difference between the median AUC for the different tidal volumes. A Kruskal Wallis test for the AUCs at the diaphram return a *p* value of 4 × 10^−4^, indicating that we reject the null hypothesis that there is no difference between the AUC for the different tidal volumes at a 1% significance level. The ANOVA table (see Table 2) provides additional test results. In Table 2, Source represents the source of the variability (i.e., Column or Error). Column represents the variability between the AUC measurements obtained for the different tidal volume groups. Error represents the variability of the AUC measurements within each tidal volume group. Total represents the total variability. SS represents the sum of squares due to each source, and df is the degree of freedom associated with each source. MS is the mean squares for each source, which is the ratio SS/df. Chi-sq represents Chi-square value.

A Kruskal Wallis test for the AUCs from the EIM location returns a *p* value of 0.55 for subject 1 and 0.0001 for Subject 2, indicating that we accept the null hypothesis that there is no difference between the median AUC for the different tidal volumes a 1% significance level. The ANOVA table (see Table 3) provides additional test results. Comparing the *p*-values in Table 1 and Table 2, we conclude that the EMG data collected at the diaphragm is more consistent than those obtained at the EIM. Therefore, we use the data collected at the diaphragm (Figure 6b and d) to create the cubic splines for predicting tidal volume from mean AUC for EMG data. The data from Figure 6b,d is summarized in Table 4. The resulting cubic spline generated from Equation (1) and Table 4 is presented in Figure 7. The cubic splines in Figure 7a,b provides estimates of tidal volume from mean AUC for EMG data collected at the diaphragm of the two subjects. After fitting a linear function to the data (see Figure 7c,d), we observe that the tidal volume varies somewhat linearly with the AUCs for the range of tidal volume used in the study. The AUC values for EMG per breathing cycle for different tidal volumes collected at the diaphragm for the male subject is higher than the female subject (see Table 4 and Figure 7).

### 4.2. Extraction of Respiration Rate

#### 4.2.1. Signature Matrix Method

In this study, we present a new method of using signature matrix [33]. The method generates a series of two-dimensional “image-like” signature matrix patterns from the sensor signals. These signature matrix patterns reflect the compressed characteristics per sampling window using the calculated probabilities as pixels of an image frame. The classification zone can be set up from a set of well-planned series of subgroups (i.e., template) to estimate the characteristic (or signature) behavior of the means of a sensor signal pattern.

For all sensor channels, a signature matrix (i.e., frame) *SIG_k_* for sample subgroup *k* is defined by:(9)$$SIG_{k}=\left[\begin{array}{ccccc} sig_{11} & sig_{12} & . & . & sig_{1N}\\ sig_{21} & sig_{22} & . & . & sig_{2N}\\ . & . & sig_{ij} & . & .\\ . & . & . & . & .\\ sig_{N1} & sig_{N2} & . & . & sig_{NN} \end{array}\right]\;$$

For a selected combination of the sample sensor signals (i.e., say channels *X* and *Y*), a frequency count can be calculated from a matching column *j* and row *i* for the corresponding zone of channels *X* and *Y*. Thus, a signature matrix is a probability map that a selected subgroup (or sample) can fall into a specific classification zone. Thus:(10)$$sig_{ij}=P\left[\begin{array}{c} \left(a\;mean\;of\;X\;belongs\;to\;class\;j\;\right)\\ \cap \left(a\;mean\;of\;Y\;belongs\;to\;class\;i\;\right) \end{array}\right]$$

#### 4.2.2. Lung Motion Signature

In this study, the signature matrix can be described as a two-dimensional probability diagram (or image map) with a 3 × 3 matrix size. The method uses a two-dimensional (image-like) signature matrix pattern from the Equation (10). The data were collected in a subgroup size of 20 data points. Thus, there will be 80 subgroups per second if the sampling frequency is 4 kHz, the sampling frequency used for the lung sensor signal monitoring. Let us define the pixel elements of the template and the measured matrix at the sampling period of *k*:(11)$$sig_{ij}^{T}\equiv \left(a\;pixel\;of\;the\;template\;matrix\right)$$
(12)$$sig_{ij}^{k}\equiv \left(a\;pixel\;of\;the\;measured\;matrix\right)$$

Figure 8 shows the 3 × 3 signature matrix formation used for the high and low frequency EMG signals. These signature matrix patterns reflect the sensor signals compressed characteristics per given data length of interest. A real-time sensor data assessment can be conducted by comparing the measured signature matrix patterns (i.e., targets) with the stored patterns (i.e., templates). The method starts with dividing the signal averages into zones. This study used a 2-sigma confidence interval to classify and allocate the EMG signals to the corresponding pixels. A combination of the signal averages from the data subgroups from two sensors finds the corresponding pixel in the signature matrix, calculates a new pixel value, and assigns the new probability value to the matrix pixel. Figure 9 shows the calculated signature matrix pixel values from the high and low frequency Sound and EMG signal values from the EIM location with 1000 mL tidal volume. The calculated 3 × 3 template signature matrices are shown as follows:(13)$$SIG_{sound}^{T}\equiv \left[\begin{array}{ccc} 0.0497 & 0.1370 & 0.0392\\ \;0.1116\; & 0.3278 & 0.0000\\ 0.0518 & 0.1436 & 0.1392 \end{array}\right]$$
(14)$$SIG_{EMG}^{T}\equiv \left[\begin{array}{ccc} 0.0119 & 0.1815 & 0.0105\\ \;0.0395\; & 0.5216 & 0.0000\\ 0.0136 & 0.1804 & 0.0411 \end{array}\right]$$

The template matrix’s data window is shown in Figure 9 (inspiration; shown in blue and green boxes).

The squared difference between the template pixel and the measured pixel is given by:(15)$$d_{ij}^{k}={\left(sig_{ij}^{T}-sig_{ij}^{k}\right)}^{2}$$

Thus, the sum of squared differences at the sampling period *k* between the template and the measured signature matrices is given by:(16)$$E_{ij}^{k}=\sum_{i=1}^{N}\sum_{j=1}^{N}{\left(sig_{ij}^{T}-sig_{ij}^{k}\right)}^{2}$$

In the study, the inspiration moment can be estimated by calculating Equation (16) using the inspiration template matrices given in Figure 9 and Equation (16). The inspiration moments (shown in red arrows) were recorded at 40 kHz subgroup sample frequency and shown in Figure 10. Please note that the inspiration moments can be identified with a low level of matching errors (blue and purple circles). Figure 11 shows the breathing cycles (i.e., inspiration moments) monitored using the template signature matrices obtained from the diaphragm location.

## 5. Discussion

This study was performed from two person’s (one male and one female) breathing cycle data for a proof-of-concept. The AUC values for EMG per breathing cycle for different tidal volumes collected at the diaphragm for the male subject is higher than the female subject. It is known that lung function is influenced by multiple factors, including age, sex/gender, and height [34,35]. We speculate that the EMG data collected at the diaphragm may also change between different individuals depending on age, sex/gender, and height. Future work would involve using data from a wide range of individuals of different ages, sex/gender, and heights to understand better how these factors affect our model predictions.

Our study shows that EMG measurements at both the EIM and diaphragm vary nonlinearly with tidal volume. The cubic spline interpolation shows that at both diaphragm and EIM, the distribution for AUC is somewhat linear for all the tidal volumes. Also, at EIM, there is a wider distribution for AUC. Compared to EIM, the AUC distribution at the diaphragm has a narrower distribution. The data collected at the diaphragm provides a better approximation of the tidal volume since AUC distribution at the diaphragm has lower variance, and the *p*-value from the Kruskal-Wallis test is smaller than the AUC distribution at the EIM. Therefore, the diaphragm’s AUC distribution is better suited for creating predictive models to estimate tidal volume. We used cubic spline interpolation to create a predictive model for tidal volume. First, we define an inverse problem where we assume that we seek the corresponding tidal volume for given AUC inputs.

The respiration rate is one of the clinically significant vital signs. Our wearable sensor system provides an exciting novel approach for continuous respiration rate monitoring using the new signature matrix method. Our approach is worth further investigation and analyses on a larger sample size since only two individuals’ breathing cycle data were used in this study. Future work would include comparing other measuring methods of the respiration rate, such as impedance measurement and respiratory inductance plethysmography [36,37,38].

## 6. Conclusions

The monitoring and analysis of respiration data remotely and continuously provide essential healthcare information for the users and the public healthcare communities, including hospitals. For example, assessment of cyclical fluctuations of heart rate, combined with the study of respiration-disorders, allows reliable recognition of the signs of illness such as coronavirus disease, etc. In particular, the coronavirus disease asks for isolation such as social distancing and quarantine and hospital capacity management to treat many people quickly. The coronavirus outbreak leads to a permanent shift in the acceptance of the technology innovations that reduce human-to-human contact and automate intelligent processes. The wearable describes a multi-sensor multi-channel wireless platform for smart remote health monitoring to address one of the fastest-growing medical research due to the increasing aging population and associated diseases.

## 7. Patents

The following patent is partially resulting from the work reported in this manuscript: Moon, K.S., Lee, S.Q. An Interactive Health-Monitoring Platform for Wearable Wireless Sensor Systems. PCT/US20/51136 (under review).

## Figures and Tables

**Figure 1 sensors-21-01393-f001:**
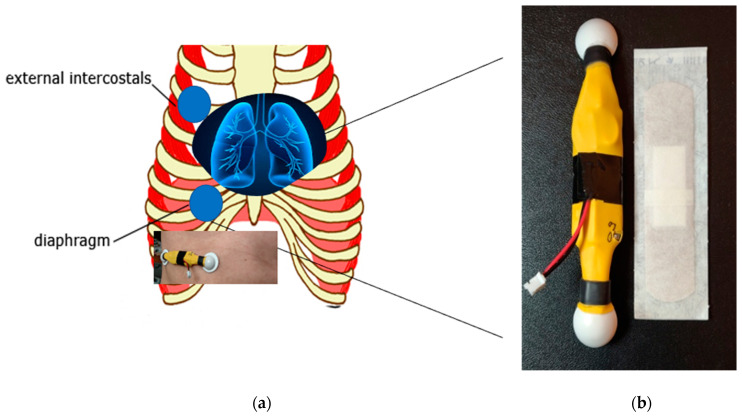
The wireless wearable health sensor: (**a**) Placements of the sensor; (**b**) The external views of the sensor.

**Figure 2 sensors-21-01393-f002:**
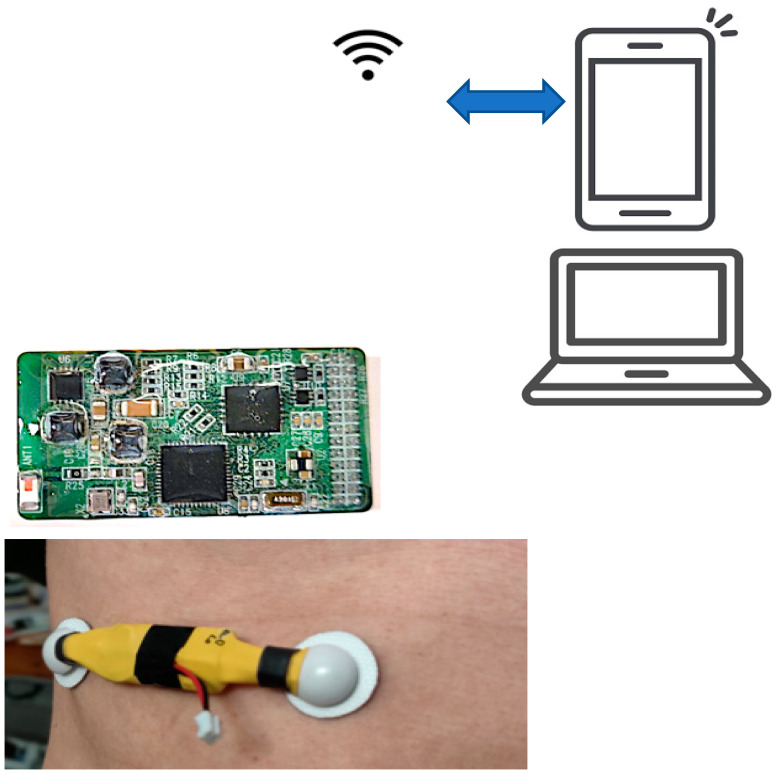
Overall system block diagram.

**Figure 3 sensors-21-01393-f003:**
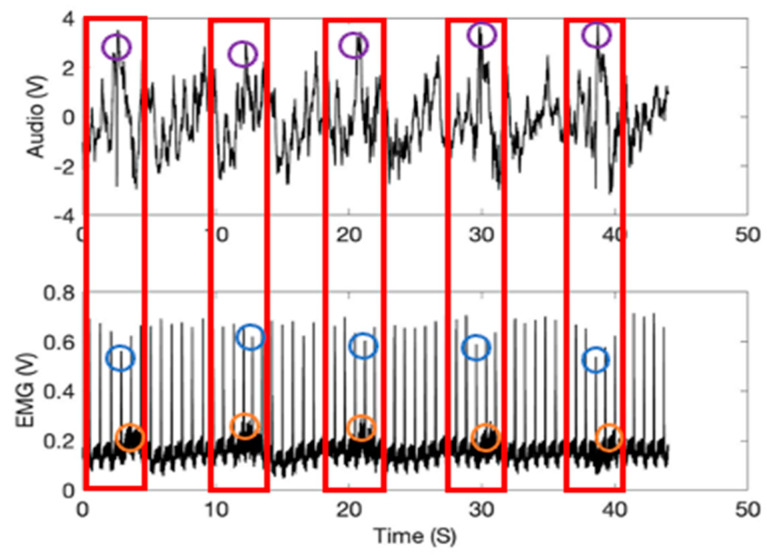
The time and amplitude characteristics of normal lung sound and EMG signals obtained from the diaphragm muscle location (1000 mL tidal volume; one breathing cycle: 4-s inspiration (red box) and 4-s expiration).

**Figure 4 sensors-21-01393-f004:**
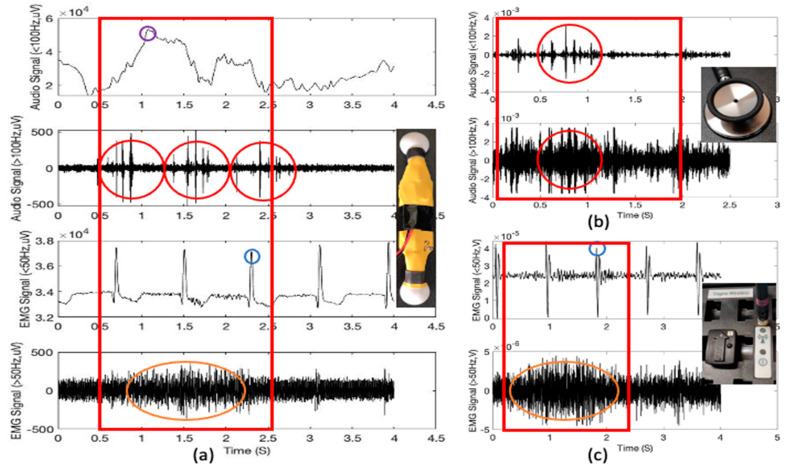
(**a**) Wearable sensor signals: The time and amplitude characteristics of typical lung sound and EMG signals obtained from diaphragm muscle location (1000 mL tidal volume; one breathing cycle: 2-s inspiration (red box) and 2-s expiration); (**b**) Reference stethoscope audio signals obtained from diaphragm muscle location (1000 mL tidal volume; 2-s inspiration (red box)); (**c**) Reference EMG signals received from diaphragm muscle location (1000 mL tidal volume; 2-s inspiration (red box)).

**Figure 5 sensors-21-01393-f005:**
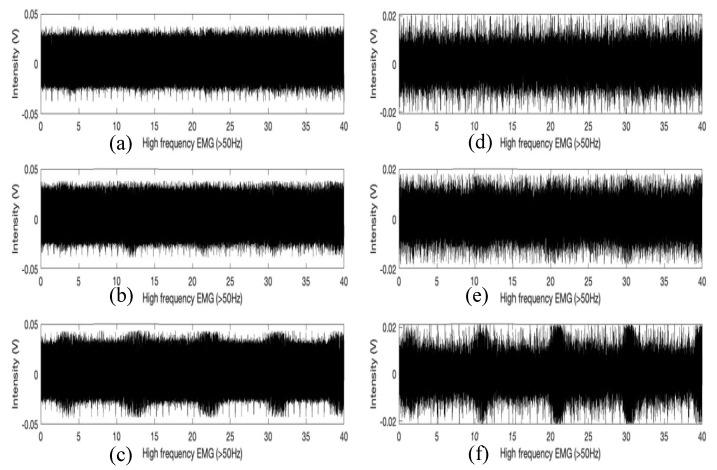
The EMG signal above 50 Hz were recorded at 4 kHz sampling frequency. The experiments were conducted to have the same 2-s inspiration, 2-s expiration, 5-s pause cycle breathing pattern from diaphragm location (**a**–**c**) and the EIM location (**d**–**f**): (**a**) 500 mL tidal volume; (**b**) 750 mL tidal volume; (**c**) 1000 mL tidal volume; (**d**) 500 mL tidal volume; (**e**) 750 mL tidal volume; (**f**) 1000 mL tidal volume.

**Figure 6 sensors-21-01393-f006:**
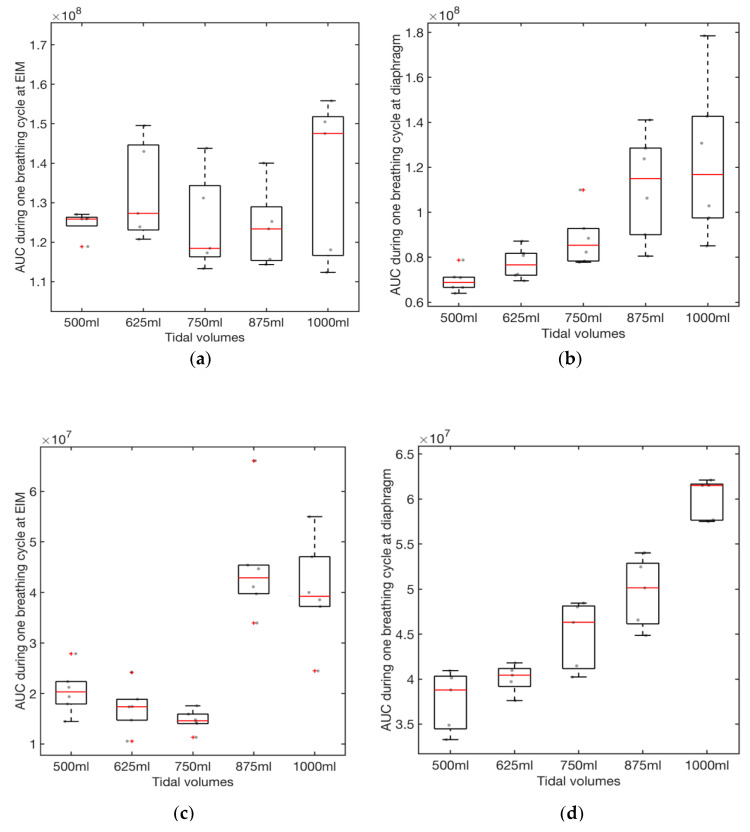
Box plots for the AUCs during one breathing cycle for different tidal volumes: (**a**) Sensor is placed at the EIM (subject 1, male); (**b**) Sensor is placed at the diaphragm (subject 1)**;** (**c**) Sensor is placed at the EIM (subject 2, female); (**d**) Sensor is placed at the diaphragm (subject 2).

**Figure 7 sensors-21-01393-f007:**
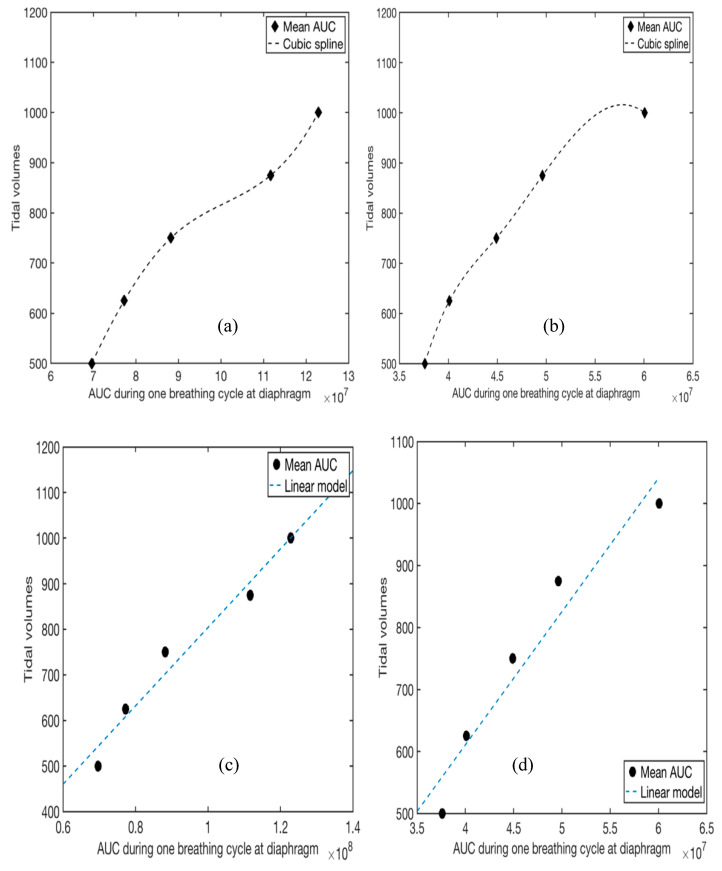
Curve fitting. The y-axis represents the tidal volume and the x-axis represents the mean AUC values calculated from the EMG data (obtained at the diaphragm) for the different tidal volume during one breathing cycle. (**a**,**b**) Cubic spline interpolation. (**a**) subject 1 (male), (**b**) subject 2 (female); (**c**) Fitting a linear model to subject 1’s data at the diaphragm yields $f\left(x\right)=8.6\times 10^{6}x-56$ as the best fit line with 95% confidence bounds (adjusted R^2^ = 0.96). Here x represents the AUC during one breathing cycle obtained at the diaphragm and f represents the tidal volume. (**d**) Fitting a linear model to subject 2’s data at the diaphragm yields $f\left(x\right)=2.2\times 10^{5}x-249.4\;$ as the best fit line with 95% confidence bounds (adjusted R^2^ = 0.92).

**Figure 8 sensors-21-01393-f008:**
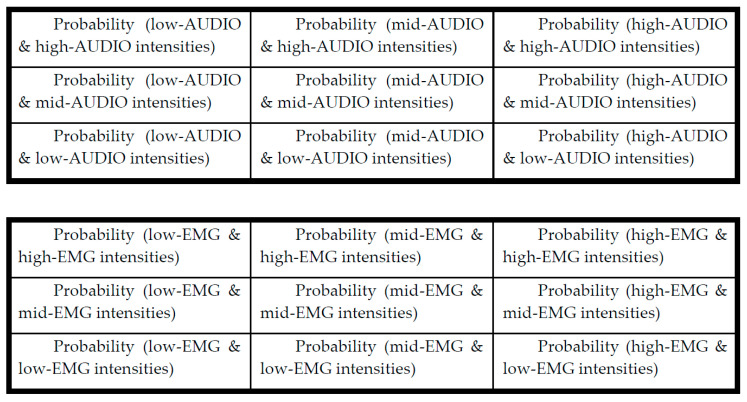
An example formation of the 3 × 3 signature matrices used for the sensor’s audio and EMG signals.

**Figure 9 sensors-21-01393-f009:**
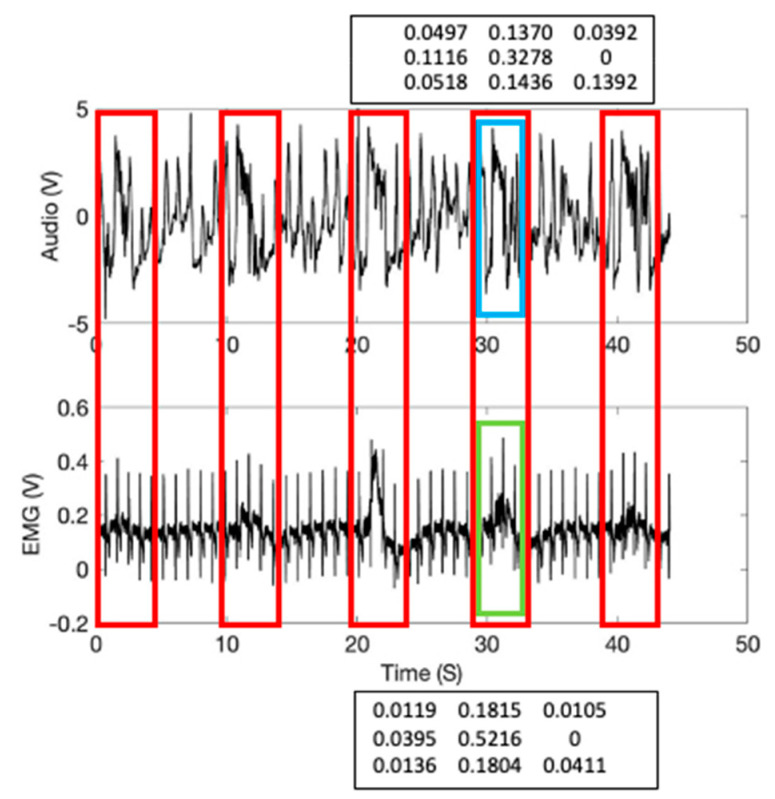
The signature matrix’s data windows in the sensor signals at EIM location (1000 mL tidal volume): Sound template window (inspiration; shown in a blue box); EMG template window (shown in a green box)**.**

**Figure 10 sensors-21-01393-f010:**
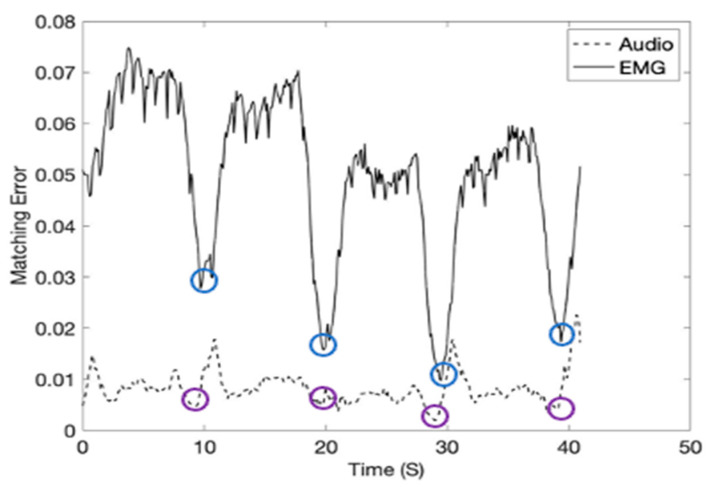
The breathing cycles (i.e., inspiration moments) monitored using the template signature matrices. The sensor signals were obtained from the EIM location (1000 mL tidal volume).

**Figure 11 sensors-21-01393-f011:**
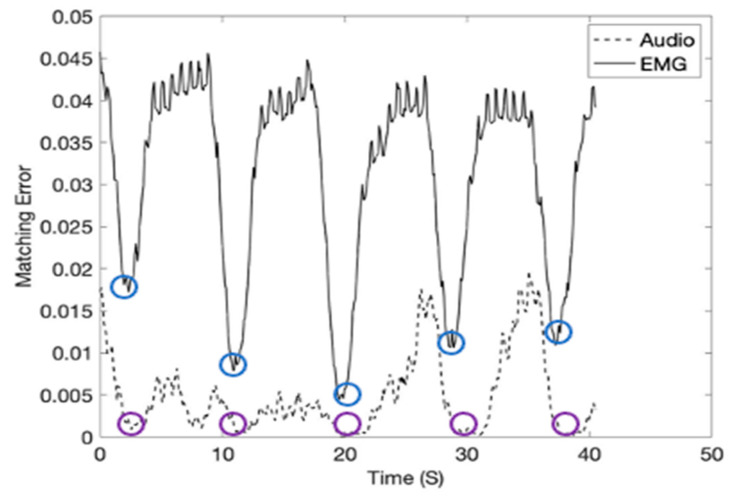
The breathing cycles (i.e., inspiration moments) monitored using the template signature matrices. The sensor signals were obtained from the diaphragm location (1000 mL tidal volume).

**Table 1 sensors-21-01393-t001:** The specifications of the wireless wearable health sensor.

Specification	Description	Value
Sensor size and weight	sensor packaging & battery (no electrodes)	(60 × 12 × 5) mm 15 g
Sensor circuit size	PCB & battery	(30 × 10 × 3) mm 6 g
Power source	rechargeable battery	8 h/charging
Data transmission	Bluetooth wireless	1 M bps in 2 m
Sound transducer	piezoelectric plate	10 mm diameter
EMG electrodes	Disposable Ag/AgCl standard pre-gelled and self-adhesive	(20 × 20) mm
Front-end circuit	Intan Tech Chip	10 mV, 16 bit, 8 ch
Onboard CPU	ARM Cortex M4	4096 Hz/ch sampling rate
Wireless circuit	NRF 52X	2.4 G ESB RF

**Table 2 sensors-21-01393-t002:** Kruskal-Wallis ANOVA table for AUC at the diaphragm.

**Subject 1**					
**Source**	**SS**	**df**	**MS**	**Chi-sq**	**Prob > Chi-sq**
Columns	1573.0	4	393.1	20.3	$4\times 10^{-4}$
Error	674.5	25	27.0		
Total	2247.5	29			
**Subject 2**					
**Source**	**SS**	**df**	**MS**	**Chi-sq**	**Prob > Chi-sq**
Columns	1095.0	4	273.9	20.2	$4\times 10^{-4}$
Error	203.9	20	10.2		
Total	1299.5	24			

**Table 3 sensors-21-01393-t003:** Kruskal-Wallis ANOVA table for AUC at EIM.

**Subject 1**					
**Source**	**SS**	**df**	**MS**	**Chi-sq**	**Prob > Chi-sq**
Columns	165.6	4	41.4	3.06	0.55
Error	1133.9	20	56.7		
Total	1299.5	24			
**Subject 2**					
**Source**	**SS**	**df**	**MS**	**Chi-sq**	**Prob > Chi-sq**
Columns	1773.7	4	443.4	22.9	0.0001
Error	473.8	25	19.0		
Total	2247.5	29			

**Table 4 sensors-21-01393-t004:** Mean AUC for EMG per breathing cycle for different tidal volumes. EMG data collected by sensors at the diaphragm.

Mean AUCs for EMG per Breathing Cycle					
Subject 1, Male			Subject 2, Female		
*j*	Mean AUC $\left(x_{j}\right)$	Tidal Volume (mL) $\left(y_{j}\right)$	*j*	Mean AUC $\left(x_{j}\right)$	Tidal Volume (mL) $\left(y_{j}\right)$
*0*	$0.70\times 10^{8}$	500	*0*	$0.70\times 10^{8}$	500
*1*	$0.77\times 10^{8}$	625	*1*	$0.40\times 10^{8}$	625
*2*	$0.88\times 10^{8}$	750	*2*	$0.45\times 10^{8}$	750
*3*	$1.12\times 10^{8}$	875	*3*	$0.50\times 10^{8}$	875
*4*	$1.23\times 10^{8}$	1000	*4*	$0.61\times 10^{8}$	1000

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
