# Peer review of "Extraction and Analysis of Respiratory Motion Using a Comprehensive Wearable Health Monitoring System"

_sensors, 2021, doi:10.3390/s21041393_

Round 1
Reviewer 1 Report
Although the approach of the measurement setup is a nice one, and might provide merits in clinical practice, the entire study pivots around data measured on only ONE single volunteer, instructed to breathe at 3 different tidal volumes (for which I did not even see a reference device on tidal flow added). This results in 3 data points in figure 7 that are then connected by a cubic spline interpolation. It is not surprising that a cubic spline interpolation can be fitted onto 3 data points...
What I have done is provide some input on the manuscript, annotated with "yellow sticky notes" in the PDF.
I have NOT gone through the entire manuscript: When I finally read that all measurements where done on only ONE volunteer, I felt it would be more useful to advise the authors to perform more measurements on additional volunteers. Typically in these early reconnaissance articles for new sensors, one would at the very, very least want to see 5 volunteers, but more is better.

Author Response
We are grateful to the reviewers for their insightful comments on our paper. We have been able to incorporate changes to reflect most of the suggestions provided by the reviewers. We have highlighted the changes in the red color within the manuscript. Please find a point-by-point response to your concerns. We hope that you find our responses satisfactory and that the paper is now acceptable for publication.
Reviewer 1:
Comments and Suggestions for Authors
1) Although the approach of the measurement setup is a nice one, and might provide merits in clinical practice, the entire study pivots around data measured on only ONE single volunteer, instructed to breathe at 3 different tidal volumes (for which I did not even see a reference device on tidal flow added). This results in 3 data points in figure 7 that are then connected by a cubic spline interpolation. It is not surprising that a cubic spline interpolation can be fitted onto 3 data points...
(Thank you for pointing this out. We agree with this comment. Therefore, we have added one more person’s experimental data with the five different tidal volumes in the revised manuscript. This change can be found in line numbers 211-260.)
2) What I have done is provide some input on the manuscript, annotated with "yellow sticky notes" in the PDF.
(Thank you for providing us your kind input. We revised the manuscript accordingly.)
3) I have NOT gone through the entire manuscript: When I finally read that all measurements where done on only ONE volunteer, I felt it would be more useful to advise the authors to perform more measurements on additional volunteers. Typically in these early reconnaissance articles for new sensors, one would at the very, very least want to see 5 volunteers, but more is better.
(Thank you for pointing this out. We agree with this comment. Therefore, we have added one more person’s experimental data with the five different tidal volumes in the revised manuscript. This change can be found in line numbers 211-260. Please understand that the IRB under COVID-19 pandemic limited including more subjects than two persons for this study.)
Reviewer 2 Report
This paper addresses the development of a wearable health monitoring device that captures both respiratory (lung sounds) and ECG/EMG signals. The system characteristics and functionality are briefly presented being the focus of the paper put on the analysis and extraction of information. For this purpose, the authors propose a model of respiration function and the joint analysis of the various captured signals to infer respiration volume and rate, which are then analyzed together the ECG signal .
The procedure for the extraction of the respiration rate should be described more in detail. The analysis of the effectiveness of this method, compared to alternative methods, should have been included.
The paper is well written but a final review of the written English is recommended.
Author Response
We are grateful to the reviewers for their insightful comments on our paper. We have been able to incorporate changes to reflect most of the suggestions provided by the reviewers. We have highlighted the changes in the red color within the manuscript. Please find a point-by-point response to your concerns. We hope that you find our responses satisfactory and that the paper is now acceptable for publication.
Reviewer 2:
Comments and Suggestions for Authors
1) This paper addresses the development of a wearable health monitoring device that captures both respiratory (lung sounds) and ECG/EMG signals. The system characteristics and functionality are briefly presented being the focus of the paper put on the analysis and extraction of information. For this purpose, the authors propose a model of respiration function and the joint analysis of the various captured signals to infer respiration volume and rate, which are then analyzed together the ECG signal. The procedure for the extraction of the respiration rate should be described more in detail.
(Thank you for pointing this out. We agree with this comment. Therefore, we have included more explanations as well as new figures in the revised manuscript. This change can be found in line numbers 293-339.)
2) The analysis of the effectiveness of this method, compared to alternative methods, should have been included.
(Thank you for pointing this out. We agree with this comment. Therefore, we have added Figures 4(b) and 4(c) from the new experimental data from reference sensor systems. Further, the additional discussion was added in the revised manuscript. This change can be found in line numbers 135-168.)
3) The paper is well written but a final review of the written English is recommended.
(Agree. We have accordingly revised the manuscript.)
Reviewer 3 Report
The advantage of the proposed ms is the tidal volume estimation technique. However this needs to be discussed with pulmonologist and some feedback from medical community can be provided.
The detailed signal processing algorithm is also provided. However there are several methods to assess the mutual registered signal processing (see e.g. 10.1016/j.physa.2018.08.146; 10.3390/app10010209). Is it possible to update proposed method to use classification portrets of the registered signal? or reduce number of registered parameters by defining synchronization metrics? So authors could show some complemetary indicator of respiratory modulation of othe physilogical signals? (for example 10.1109/SCM.2015.7190446)
Author Response
We are grateful to the reviewers for their insightful comments on our paper. We have been able to incorporate changes to reflect most of the suggestions provided by the reviewers. We have highlighted the changes in the red color within the manuscript. Please find a point-by-point response to your concerns. We hope that you find our responses satisfactory and that the paper is now acceptable for publication.
Reviewer 3:
Comments and Suggestions for Authors
1) The advantage of the proposed ms is the tidal volume estimation technique. However this needs to be discussed with pulmonologist and some feedback from medical community can be provided.
(Thank you for pointing this out. We agree with this comment. The first author, Dr. George’s research includes pulmonary/medical area, and the tidal volume estimation was reviewed with her colleague. We have further added one more person’s experimental data with the five different tidal volumes in the revised manuscript. This change can be found in line numbers 211-260.)
2) The detailed signal processing algorithm is also provided. However there are several methods to assess the mutual registered signal processing (see e.g. 10.1016/j.physa.2018.08.146; 10.3390/app10010209). Is it possible to update proposed method to use classification portrets of the registered signal? or reduce number of registered parameters by defining synchronization metrics? So authors could show some complemetary indicator of respiratory modulation of othe physilogical signals? (for example 10.1109/SCM.2015.7190446
(Thank you for pointing this out. We agree with this comment. However, we would like to include your suggestion for our future study due to the time commitments. We appreciate your kind understanding.)
Reviewer 4 Report
The authors describe developemnt of a wearable health monitoring system. The technical descrption of the developed device is almost complete.
I miss the information about the sampling frequency of ECG and EMG signals.
My major comment concerns missing comparison with a calibrated medical device. It is a standard procedure that a newly developed sensor or device must be compared with an already approved device. I understand that the authors present a prototype and its functionality. However, there is the question how they can be sure that the measured data and signals are correct.
There is a mistake in figure 4 legend or at least contradiction with the text on page 5, line 142 and figure 4 itself (part d). In the legend there is (d) EMG signal and in the text and figure there is ECG signal, which is presumably correct. Please correct the legend.
Figure 5: Since the x-axis (time) is too squeezed, the figure is not readable. There are only black stripes with some peaks. Please try to adjust the figure so that it brings some information to the reader. The texts at x- and y-axes are not readable.
English language is fine. There are some mistyping errors, e.g. "spine" instead of "spline".
Author Response
We are grateful to the reviewers for their insightful comments on our paper. We have been able to incorporate changes to reflect most of the suggestions provided by the reviewers. We have highlighted the changes in the red color within the manuscript. Please find a point-by-point response to your concerns. We hope that you find our responses satisfactory and that the paper is now acceptable for publication.
Reviewer 4:
Comments and Suggestions for Authors
1) The authors describe development of a wearable health monitoring system. The technical description of the developed device is almost complete. I miss the information about the sampling frequency of ECG and EMG signals.
(Thank you for pointing this out. We agree with this comment. The sample rate is 4k Hz per channel. This information can be found in line numbers 104, 120, and 283.)
2) My major comment concerns missing comparison with a calibrated medical device. It is a standard procedure that a newly developed sensor or device must be compared with an already approved device. I understand that the authors present a prototype and its functionality. However, there is the question how they can be sure that the measured data and signals are correct.
(Thank you for pointing this out. We agree with this comment. Therefore, we have added Figures 4(b) and 4(c) from the new experimental data from reference sensor systems. Further, the additional discussion was added in the revised manuscript. This change can be found in line numbers 135-168.)
3) There is a mistake in figure 4 legend or at least contradiction with the text on page 5, line 142 and figure 4 itself (part d). In the legend there is (d) EMG signal and in the text and figure there is ECG signal, which is presumably correct. Please correct the legend.
(Agree. We have accordingly revised the figure.)
4) Figure 5: Since the x-axis (time) is too squeezed, the figure is not readable. There are only black stripes with some peaks. Please try to adjust the figure so that it brings some information to the reader. The texts at x- and y-axes are not readable.
(Agree. We have accordingly revised the figure.)
English language is fine. There are some mistyping errors, e.g. "spine" instead of "spline".
(Agree. We have accordingly revised the sentence.)
Reviewer 5 Report
The topic is quite interesting. However, the consideration of just one subject is quite limiting. The first three figures are hardly readable. Conclusions must be revised pointing out the strong limitation of one subject and by introducing some statistical considerations.
Author Response
We are grateful to the reviewers for their insightful comments on our paper. We have been able to incorporate changes to reflect most of the suggestions provided by the reviewers. We have highlighted the changes in the red color within the manuscript. Please find a point-by-point response to your concerns. We hope that you find our responses satisfactory and that the paper is now acceptable for publication.
Reviewer 5:
Comments and Suggestions for Authors
1) The topic is quite interesting. However, the consideration of just one subject is quite limiting. The first three figures are hardly readable. Conclusions must be revised pointing out the strong limitation of one subject and by introducing some statistical considerations
(Thank you for pointing this out. We agree with this comment. Therefore, we have added one more person’s experimental data with the five different tidal volumes in the revised manuscript. This change can be found in line numbers 211-260. Please understand that the IRB under COVID-19 pandemic limited including more subjects than two persons for this study.)
Round 2
Reviewer 1 Report
Lines 22-23: "...area under the curve (AUC) values were calculated against time per breathing cycle from the EMG graphs measured on the diaphragm as well as on the EIM." Do the AUC values come from the spirometer? If so, clarify. If not, then what is meant here?
Lines 23-24: "The calculated AUC values represent the expansion of the diaphragm as well as EIM." This sentence is fully unclear. Sharply formulate what is meant. Are you looking at correlation between spirometer and EIM?
For the claim of boldly estimating tidal lung volumes, the data is too thin (sorry). Lines 27-28: Would suggest to replace "Our findings show that the new sensor can be used to estimate tidal lung volume from EMG measurements obtained at the diaphragm." Instead may be better: "Our findings show that the new sensor can be used to measure respiration rate and variations thereof, and holds potential to estimate tidal lung volume from EMG measurements obtained from the diaphragm." This honors the nice results achieved on the acoustic part, while not overinflating the conclusions from very limited data.
Line 61: "This paper presents a noble way to monitor" probably you mean "novel" (recall also highlighting this in first review).
Lines 116-118: In your rebuttal letter and the abstract, you indicate to have expanded the number of tidal volumes from 3 to 5 and the number of volunteers from 1 to 2. Yet, in line 116-118 you still mention 3 tidal volumes and one volunteer. How thorough has your manuscript rewriting been?
Line 118: The reference devices is called "a spirometer" without stating model and manufacturer. Your reference device is a crucial part of the story, it should be described properly with at least adding these details.
Figure 7: The figure has a double caption, one in the figure itself and one as caption text. The caption inside the figure says "AUC during one breathing cycle at diaphragm". That would not make much sense?! You measure at only 2 volunteers, at 5 tidal volumes. The only thing you have plentiful is the number of breaths, so distill data from that plentitude of breaths, label the points for both volunteers differently and put that in a graph.
Honestly, I do like the idea with simultaneous audio and EMG measurement. The audio data also seems to produce pretty useful respiration rate data. But why is there so much emphasis on a a complicated claim of estimating tidal volume from EMG measurements, based upon a very low number of people?
Am providing this feedback truly well-intended, but think it a waste of time to further complete the review (sorry).
Author Response
We are grateful again for your comments on our paper. We have been able to incorporate additional changes to reflect most of the additional suggestions provided by the reviewers. We have highlighted the changes in the yellow color within the manuscript. Please find a point-by-point response to your concerns. We hope that you find our responses satisfactory and that the paper is now acceptable for publication.
Lines 22-23: "...area under the curve (AUC) values were calculated against time per breathing cycle from the EMG graphs measured on the diaphragm as well as on the EIM." Do the AUC values come from the spirometer? If so, clarify. If not, then what is meant here?
(Response: Thank you for pointing this out. We agree with this comment. We have addressed this comment by adding the following text on lines 21-25.
The tidal volumes were controlled with a spirometer. The duration of each breathing cycle was 8 seconds and was timed using a metronome. For each of the different tidal volumes, the EMG data was plotted against time and the area under the curve (AUC) was calculated.)
Lines 23-24: "The calculated AUC values represent the expansion of the diaphragm as well as EIM." This sentence is fully unclear. Sharply formulate what is meant. Are you looking at correlation between spirometer and EIM?
(Response: Thank you for providing us your kind input. We have addressed this comment by adding the following text on lines 24-25.
The AUC calculated from EMG data obtained at the diaphragm and EIM represent the expansion of the diaphragm and EIM respectively.)
For the claim of boldly estimating tidal lung volumes, the data is too thin (sorry). Lines 27-28: Would suggest replacing "Our findings show that the new sensor can be used to estimate tidal lung volume from EMG measurements obtained at the diaphragm." Instead may be better: "Our findings show that the new sensor can be used to measure respiration rate and variations thereof and holds potential to estimate tidal lung volume from EMG measurements obtained from the diaphragm." This honors the nice results achieved on the acoustic part, while not overinflating the conclusions from very limited data.
(Response: We agree with your comment and have replaced the sentence on lines 28-30.
``Our findings show that the new sensor can be used to estimate tidal lung volume from EMG measurements obtained at the diaphragm”
with
``Our findings show that the new sensor can be used to measure respiration rate and variations thereof and holds potential to estimate tidal lung volume from EMG measurements obtained from the diaphragm”)
Line 61: "This paper presents a noble way to monitor" probably you mean "novel" (recall also highlighting this in first review).
(Response: Thank you for bringing this to our attention. We have corrected this. We used ‘noble’ instead of ‘novel’ about three times and have corrected this in the manuscript. )
Lines 116-118: In your rebuttal letter and the abstract, you indicate to have expanded the number of tidal volumes from 3 to 5 and the number of volunteers from 1 to 2. Yet, in line 116-118 you still mention 3 tidal volumes and one volunteer. How thorough has your manuscript rewriting been?
(Response: We are sorry, this was a typo. We have addressed this comment by adding the following text on lines 131-135
``In the study, a set of experimental sensor signals was collected from a healthy male and female with slow respiration with a constant rate (i.e., 4-second inspiration and 4-second expiration). The breathing cycle was controlled using a metronome (https://www.imusic-school.com/en/tools/online-metronome/). Five different tidal volumes (i.e., 1000ml, 875ml, 750 ml, 625ml and 500ml) were controlled by using a Voldyne 5000 Spirometer (Hudson RCI, calibrated with PF100 digital Peak Flow & FEV1 Meter, Microlife).’’)
Line 118: The reference devices is called "a spirometer" without stating model and manufacturer. Your reference device is a crucial part of the story, it should be described properly with at least adding these details.
(Response: Agree. We have addressed this comment in our response to the preceding comment.)
Figure 7: The figure has a double caption, one in the figure itself and one as caption text. The caption inside the figure says "AUC during one breathing cycle at diaphragm". That would not make much sense?! You measure at only 2 volunteers, at 5 tidal volumes. The only thing you have plentiful is the number of breaths, so distill data from that plentitude of breaths, label the points for both volunteers differently and put that in a graph.
Honestly, I do like the idea with simultaneous audio and EMG measurement. The audio data also seems to produce pretty useful respiration rate data. But why is there so much emphasis on a a complicated claim of estimating tidal volume from EMG measurements, based upon a very low number of people?
Am providing this feedback truly well-intended, but think it a waste of time to further complete the review (sorry).
(Response: Thank you for pointing this out. We agree with this comment. We have edited Figure 7 caption to clarify that the x-axis represents the mean AUC values calculated from the EMG data (obtained at the diaphragm) for the different tidal volume during one breathing cycle. We have presented the graphs for both volunteers. We are sorry that we could only use 2 volunteers. It is currently very risky to collect respiratory information from multiple subjects because of COVID-19 pandemic. It would be very difficult to obtain IRB approval to include multiple subjects because of the pandemic. Both subject 1 and 2 are coauthors on the paper.
We believe that being able to predict the tidal volume from EMG data is vital and has the potential to enhance the assessment of lung function in diseases such as COPD. In particular, if the sensor and math/predictive model is tested on a larger sample size and standardized based on race, sex, height etc., it could potentially enhance the early detection of COPD by spotting abnormalities in AUC versus tidal volume trend. We agree with you that we need data from multiple subjects and we will be collecting data from multiple subjects to standardize the math model based on race, sex, height etc. once the pandemic is over.
We are thankful for your feedback and suggestions for improving our manuscript. Your suggestions help improve the readability of the manuscript.
We have modified the Figure 7 caption as follows:
Curve fitting. The y-axis represents the tidal volume and the x-axis represents the mean AUC calculated from the EMG data during one breathing cycle for each tidal volume (obtained at the diaphragm). (a) - (b) Cubic spline interpolation. (a) subject 1 (male), (b) subject 2 (female); (c) Fitting a linear model to subject 1’s data at the diaphragm yields f(x)=8.6×〖10〗^6 x-56 as the best fit line with 95% confidence bounds (adjusted R2 = 0.96). Here x represents the AUC during one breathing cycle obtained at the diaphragm and f represents the tidal volume. (d) Fitting a linear model to subject 2’s data at the diaphragm yields f(x)=2.2×〖10〗^5 x-249.4 as the best fit line with 95% confidence bounds (adjusted R2 = 0.92).)
Reviewer 2 Report
The issues raised in the previous review were mostly addressed in the new version of the paper. All the process is now more clear but the authors should make another effort to obatian a better version.
It seems that the formulation of the 3x3 signature matrix presented in figure 8 is not correct as only EMG intesities are referred. Please, differentiate the two EMG intensities referred in each cell.
The written English is better but one cal still identify several flaws, such as:
- bioimpedance is written in two different forms
- the units should be separated from the numeral
- The sentence "The filter section is built with a bandpass filter program with the most heart and lung signals in the system." is not clear.
- "Connects" in "The cubic splines ??(?) connects adjacent " should be "connect"
- Calculate in "calculate probabilities as pixels of an image frame" shouuld be "calculated"
- In "This study used a 2-sima confidence" sigma is misspelled
- "calculates" in "Figure 9 shows the calculates signature matrix pixel values" should be "calculated"
The quality of the images, particularly in figure 2, should be improved.
Author Response
We are grateful again your comments on our paper. We have been able to incorporate additional changes to reflect most of the additional suggestions provided by the reviewers. We have highlighted the changes in the yellow color within the manuscript. Please find a point-by-point response to your concerns. We hope that you find our responses satisfactory and that the paper is now acceptable for publication.
The issues raised in the previous review were mostly addressed in the new version of the paper. All the process is now more clear but the authors should make another effort to obatian a better version.
It seems that the formulation of the 3x3 signature matrix presented in figure 8 is not correct as only EMG intesities are referred. Please, differentiate the two EMG intensities referred in each cell.
(Response: Thank you for pointing this out. We agree with this comment. We have addressed this comment by adding an additional 3x3 signature matrix for Audio signal. Also, we have edited Figure 8 caption to clarify that the two 3x3 signature matrices used for the sensor's Audio and EMG signals.)
The written English is better but one cal still identify several flaws, such as:
- bioimpedance is written in two different forms
- the units should be separated from the numeral
- "Connects" in "The cubic splines ??(?) connects adjacent " should be "connect"
- Calculate in "calculate probabilities as pixels of an image frame" shouuld be "calculated"
- In "This study used a 2-sima confidence" sigma is misspelled
- "calculates" in "Figure 9 shows the calculates signature matrix pixel values" should be "calculated"
(Response: Thank you for bringing this to our attention. We have corrected these in the manuscript.)
- The sentence "The filter section is built with a bandpass filter program with the most heart and lung signals in the system." is not clear.
(Response: Thank you for bringing this to our attention. We have replaced the sentence on lines 123-124.
“The wearable system has a bandpass filter program installed to maximize the sensor's heart and lung signals.”)
The quality of the images, particularly in figure 2, should be improved.
(Response: Thank you for pointing this out. We agree with this comment. We have edited Figure 2.)
Reviewer 4 Report
The authors significantly improved the manuscript. They should stress that they present a proof of concept since the experiment was performed only on one subject. So in this case we cannot speak about any quantitative evaluation.
Author Response
We are grateful again your comments on our paper. We have been able to incorporate additional changes to reflect most of the additional suggestions provided by the reviewers. We have highlighted the changes in the yellow color within the manuscript. Please find a point-by-point response to your concerns. We hope that you find our responses satisfactory and that the paper is now acceptable for publication.
The authors significantly improved the manuscript. They should stress that they present a proof of concept since the experiment was performed only on one subject. So in this case we cannot speak about any quantitative evaluation
(Response: Thank you for providing us your kind input. We have addressed this comment by adding the following text on lines 351-352.
“This study was performed from two person's (one male and one female) breathing cycle data for a proof-of-concept.”
Further, we have edited Figure 7, and we have presented the graphs for both volunteers. We are sorry that we could only use 2 volunteers. It is currently very risky to collect respiratory information from multiple subjects because of COVID-19 pandemic. It would be very difficult to obtain IRB approval to include multiple subjects because of the pandemic. Both subject 1 and 2 are coauthors on the paper.)